# Sphingosine-1-Phosphate in the Tumor Microenvironment: A Signaling Hub Regulating Cancer Hallmarks

**DOI:** 10.3390/cells9020337

**Published:** 2020-02-01

**Authors:** Laura Riboni, Loubna Abdel Hadi, Stefania Elena Navone, Laura Guarnaccia, Rolando Campanella, Giovanni Marfia

**Affiliations:** 1Department of Medical Biotechnology and Translational Medicine, LITA-Segrate, University of Milan, via Fratelli Cervi, 93, 20090 Segrate, Milan, Italy; 2Laboratory of Experimental Neurosurgery and Cell Therapy, Neurosurgery Unit, Fondazione IRCCS Ca’ Granda Ospedale Maggiore Policlinico, via Francesco Sforza 35, 20122 Milan, Italylaura.guarnaccia@policlinico.mi.it (L.G.);; 3Department of Clinical Sciences and Community Health, University of Milan, 20100 Milan, Italy

**Keywords:** sphingosine-1-phosphate, cancer, tumor microenvironment, glioblastoma, lysophospholipids, cancer hallmarks

## Abstract

As a key hub of malignant properties, the cancer microenvironment plays a crucial role intimately connected to tumor properties. Accumulating evidence supports that the lysophospholipid sphingosine-1-phosphate acts as a key signal in the cancer extracellular milieu. In this review, we have a particular focus on glioblastoma, representative of a highly aggressive and deleterious neoplasm in humans. First, we highlight recent advances and emerging concepts for how tumor cells and different recruited normal cells contribute to the sphingosine-1-phosphate enrichment in the cancer microenvironment. Then, we describe and discuss how sphingosine-1-phosphate signaling contributes to favor cancer hallmarks including enhancement of proliferation, stemness, invasion, death resistance, angiogenesis, immune evasion and, possibly, aberrant metabolism. We also discuss the potential of how sphingosine-1-phosphate control mechanisms are coordinated across distinct cancer microenvironments. Further progress in understanding the role of S1P signaling in cancer will depend crucially on increasing knowledge of its participation in the tumor microenvironment.

## 1. Introduction

Sphingosine-1-phosphate (S1P) represents the simplest natural phosphosphingolipid, and consists of a single long-chain sphingoid base linked to a phosphate group. Despite its relative simplicity, and role as intermediate metabolite in the catabolism of complex sphingolipids, S1P exhibits an extraordinary array of cell functional properties strictly related to cancer. Indeed, S1P is an important signaling molecule, able to stimulate proliferation, motility, migration, and survival of many cell types, both normal and malignant [1]. The multiplicity of S1P roles are mainly exerted through its unique inside-out signaling, via a family of five S1P-specific G protein-coupled receptors (named S1P_1_–S1P_5_) [2], differentially expressed in different cell types. In addition, S1P can also act as an intracellular signal, targeting different cellular proteins, and regulating histone acetylation and transcription [1,3,4].

Aberrant S1P metabolism, receptor expression, and signaling emerged as relevant contributors to initiation, progression, and malignant properties of different tumors [5,6,7,8], and, among them, glioblastoma (GBM) [9,10,11], one of the most aggressive neoplasms in humans.

After introducing the key features of S1P signaling and of GBM, we here review our current understanding of how different tumor and normal cells in cancer niches contribute to alterations of S1P homeostasis and signaling, and impact cancer hallmarks, progression and properties. We here focus on the emerging fields relevant to GBM as representative of highly malignant and dreaded form of cancer.

## 2. S1P Metabolism and Export

S1P homeostasis is tightly regulated by the balance between its synthesis and degradation in a metabolic process orchestrated by several metabolic enzymes [1,12]. The biosynthetic precursor for the synthesis of S1P is sphingosine, derived from the hydrolysis of ceramide during the degradation of sphingomyelin and glycosphingolipids [13]. S1P is then formed through the transfer of the γ-phosphate from adenosine triphosphate (ATP) to sphingosine, catalyzed by two isoenzymes of sphingosine kinase (SphK), namely SphK1 and SphK2. Both SphKs are subject to complex temporal and spatial regulation by multiple mechanisms, including transcriptional, translational, and post-translational ones, as well as subcellular and extracellular localization (see [14,15,16] for recent reviews).

Each SphK isozyme has variant isoforms differing only at the N-terminus [14]. Despite the fact that SphKs catalyze the formation of S1P, their product may have not only overlapping but also divergent functions within the cell, with these differences dictated also by their differential subcellular localization [17,18,19,20]. SphK1 is mainly cytoplasmic, whereas SphK2 localizes to several cell compartments depending on the cell type [21]. In response to different stimuli, both isoenzymes can acutely associate with different cell membranes, such as plasma membrane, mitochondrial-associated membranes, endosomal vesicles and phagosomes [22]. Moreover, although most S1P is synthesized intracellularly, its production may partially occur extracellularly through SphKs secreted by some cell types. Extracellular export of SphK1 was reported in vascular endothelial and HEK293 cells [23,24], and pre-apoptotic cells extracellularly release a SphK2 fragment which is enzymatically active [25].

Recently, considerable attention has been given to the role of SphKs in cancer, and of SphK1 in particular. In multiple types of cancers, elevated expression of SphK1 occurs, and the following production of S1P promotes cell survival, growth, and invasiveness [7,14,26]. In addition, a role for SphK2 in cancer emerged from studies demonstrating that high expression of SphK2 in human cancers is linked to poor patient prognosis [27].

S1P catabolism involves two major, alternative reactions. One is the irreversible cleavage of S1P to form hexadecenal and ethanolamine phosphate. This degradation reaction represents the last step in sphingolipid catabolism, catalyzed by S1P lyase, an enzyme located in the membrane of the endoplasmic reticulum (ER), with the catalytic domain facing the cytosol [28]. Alternatively, S1P can be dephosphorylated back to sphingosine, through either two specific, ER-localized S1P phosphatases (SPP1 and SPP2), or by three broad-specificity, plasma-membrane-located lipid phosphate phosphatases (LPP1-3) [29,30].

It was more than 20 years ago that intracellular S1P metabolism, including its rapid enzymatic interconversion to and from ceramide, gave rise to the sphingolipid rheostat model, whereby the balance between S1P (pro-proliferative, pro-survival) and ceramide (anti-proliferative, pro-differentiating, pro-apoptotic) exerts a crucial influence over normal and cancer cell fate [31,32]. This model has been substantiated in many reports on cancer investigations, and, along with time and the increased knowledge on the complexity of S1P signaling, it has been recently updated with the inclusion of the “inside-out” signaling by the S1P/S1P receptor (S1P1-5) axis [20]. Indeed, crucial in S1P function is its extracellular export, which occurs from different but not all cell types after intracellular generation. The cellular efflux of S1P into the extracellular environment can be mediated by several transport proteins, such as different members of the ATP-binding cassette transporters (ABCTs), Spinster 2 (Spns2) [33,34,35], as well as the major facilitator superfamily domain-containing 2b (Mfsd2b), recently identified in red blood cells [36]. Due to the efficient export by red blood cells and endothelial cells (ECs), relatively high concentrations of S1P compared with solid tissues are present in plasma [37], creating a gradient of high S1P levels in blood vs. tissues, functionally important for S1P biological properties.

## 3. Glioblastoma

Glioblastoma (GBM) is the most common type of malignant primary brain tumor in adults typically presenting as primary GBM [38]. This cancer represents a unique clinical challenge as it has strong resistance to traditional therapies, it can spread aggressively to other areas of the brain, and it may occur in susceptible areas where treatments easily cause damage to adjacent healthy tissue. In addition, the tumor itself is characterized by heterogeneous areas of necrotizing tissue and peritumoral oedema, and are often asymptomatic until reaching a large size [39]. The deleterious malignancy of GBM is exemplified by the median survival of its patients, which is only about three months without prompt treatment [40]. Current GBM therapies include surgery, radiation therapy, and chemotherapy. However, even with these therapies, the average survival of GBM patients after diagnosis remains as short as 12–15 months [41,42], and only 3–5% of patients survive more than three years [43]. GBM recurrence is nearly a rule, and, under optimal therapy, recurrent GBM patients exhibit a median survival of only 5–7 months [44]. Individual tumor cells can be found far from the primary tumor site, often crossing great distances into the contralateral hemisphere [45]. These cells cannot be isolated for surgical resection, or easily targeted by irradiation, and thus represent sources for tumor recurrences [46]. In an attempt to address residual tumor cells, Temozolomide (TMZ) is included as adjuvant chemotherapy in the current standard of care.

In bright contrast to the relevant progress achieved in terms of outcomes for different cancers, and to the rapidly growing knowledge of the molecular pathogenesis of GBM, GBM remains a serious threat and is widely incurable [47]. Indeed, despite current advances in multimodal therapies, including advanced surgery, radiotherapy, and chemotherapy, and the development of innovative, targeted therapies, the outcome for patients with GBM is nearly always fatal [48]. Various obstacles hamper development of effective therapies, including high proliferation rate, intensive angiogenesis, pervasive tumor cell infiltration, therapeutic resistance, both cellular and molecular heterogeneity, and not least the lack of a full understanding of the pathobiology of the disease.

## 4. S1P Level and Metabolism in GBM

Studies performed on human samples demonstrated that a significant decrease of the ceramide content in human GBM tissues [49] is associated to a marked increase of S1P level, accounting for about an order of magnitude [50]. Consequently, a highly significant imbalance of the sphingolipid rheostat occurs, shifting the S1P/ceramide ratio toward S1P by about 20-fold. Moreover, since different ceramide molecular species exist and execute distinct functions [51], it is relevant that all GBM had a lower content of the C18 ceramide [50], a species recognized as crucial in promoting cancer cell death [52]. Noticeably, the S1P/ceramide shift increases with malignancy in tumor tissues and is directly related to patient survival, indicating that this imbalance is common and represents a similarity in GBM, independent of the heterogeneous genetic and molecular fingerprint of this cancer. Using a combination of liquid chromatography and tandem mass spectrometry, differences in the amounts of cellular S1P were found in human glioma cell lines [53], suggesting heterogeneity of S1P metabolism and/or export in GBM cells.

To maintain high levels of S1P, GBM exhibits multiple aberrations in S1P metabolism, with different enzymes contributing to the S1P upregulation. Indeed, different studies indicate that GBM malignancy is associated with both an increased drive of the cellular pathway that converts ceramide to S1P, and a decrease of the pathways mediating S1P removal (Figure 1). A key point appears to reside in the SphK-mediated synthesis of S1P, as substantiated by different findings demonstrating that inhibition of SphKs (particularly of SphK1) results in the reduction of different S1P-mediated effects (see below). In particular, several studies reported that human GBM displays a high expression level of SphK1, and SphK1 mRNA positively correlates with S1P level [50,54,55,56,57]. High SphK1 expression has been found to be correlated with a significant poor prognosis in patients with GBM by some studies [54,55], but such a relation was not detected in more recent reports [50,57,58]. Despite these contradictory findings in tumor heterogeneity, the mechanisms underlying SphK1 gene overexpression in GBM remain unclarified. An intriguing possibility resides in the very recent findings showing that the long non-coding RNA named *Khps1* (as it is transcribed in antisense orientation to the *SphK1*) induces *SphK1* expression via recruitment of the transcription factor E2F1 to the *SphK1* promoter [59]. This hypothesis appears plausible on the base of recent evidence showing that long non-coding RNAs are key players in GBM pathogenesis [60], and E2F1 acts as a common regulator of differentially expressed genes in GBM, despite its genetic heterogeneity [61]. Opposite findings were also reported for SphK2 expression in GBM. In contrast to SphK1, Abuhusain et al. [50] reported that SphK2 expression in GBM tissues was 3-fold lower than in normal grey matter. On the contrary, Quint et al. [56] found that the mRNA expression of SphK2 in primary GBM was 25-fold higher than in normal brain and this enzyme expression decreases in both recurrent and secondary GBMs. The reason for these opposite findings is at present unclear. Noting that notwithstanding each SphK isoenzyme has variant isoforms differing only at the N-terminus [14], the vast majority of the reported studies on SphK expression in GBM do not specify the targeted specific isoform of the enzyme. Indeed, different unique isoforms of the human SphK1, differing at the N-terminus (hSphK1a-c) [24,62] and with different intrinsic properties [63], have been identified. In addition, the SphK2 gene encodes different predicted N-terminal-extended variants [64] that remain poorly investigated to date. The best-characterized variant is the short isoform (SphK2-S), which represents the most investigated one in the literature. The large isoform (SphK2-L) is not expressed in rodents, but appears the predominant form in several human cell lines and tissues, and thus more important in humans [64].

Functional to the high expression of SphKs is the availability of sphingosine, controlled by the interconversion of ceramide and sphingosine. The shift from ceramide to S1P increases with increasing glioma cancer grade [50]. It has been reported that a higher S1P/ceramide ratio contributes to a higher recurrence rate, implying the S1P signaling is a potent therapeutic target for the treatment of GBM [65]. A recent paper reported that Bcl2L13, the atypical member of the Bcl-2 family overexpressed in GBM, inhibits ceramide synthase [66]. This would likely result in the reduction of the salvage pathway for complex sphingolipid biosynthesis [67], and in facilitating sphingosine use by SphKs. In addition, the acid ceramidase was found significantly upregulated in GBM specimens, particularly in CD133^+^ GBM stem cells (GSCs), and was associated with poor GBM patient survival [50,68,69]. Besides reducing ceramide, the variations (in opposite directions) of ceramide synthase and acid ceramidase (Figure 1) appear to concur in favoring the availability of sphingosine as a substrate for SphKs, and thus the overproduction of S1P in GBM.

In addition to SphK variations, two enzymes involved in S1P degradation are altered in GBM, further potentiating the metabolic events leading to high levels of S1P in this cancer. First, it was found that the chromosomal region containing the gene for S1P lyase is deleted in human GBMs [70], suggesting that S1P upregulation is also favored by a reduction of its catabolism. Second, the S1P phosphatase 2 (hSPP2), an S1P-specific phosphohydrolase localized to the ER [71], is significantly downregulated in GBMs, its expression being inversely related to S1P levels and associated with poor patient survival [50], most likely impairing sphingosine recycling to ceramide at the ER. Consistently, it was reported that a preferential channeling of sphingosine formed in the lysosomes into S1P synthesis occurs in GBM cells, whereas S1P is mainly recycled into ceramide in neurons, astrocytes, and oligodendrocytes [72,73].

Noticeably, the imbalance of the sphingolipid rheostat and related metabolic enzymes was also reported in low-grade gliomas, which have a relatively good prognosis and prolonged survival in comparison with GBM. The elevation of the S1P/ceramide ratio was found less pronounced in low-grade than high-grade malignant tumors [49,50], revealing that this variation is not only common to all glioma tissue compared to normal brain, but also that its magnitude is directly related to glioma malignancy. Interestingly, similarly to gliomas, human prostate cancer was reported to exhibit an increased drive to maintain high S1P levels, with opposite variations of SphK1 (upregulated) and S1P lyase (downregulated) versus normal tissue, which were all significantly associated with tumor grade and aggressiveness [74]. Moreover, a high S1P/ceramide ratio, associated with increased expression of SphK1 and reduced alkaline sphingomyelinase, was found directly related to high malignancy potential of colorectal cancer [75]. In addition, elevated S1P levels were directly related to SphK1 expression as well as to high malignant grade in human breast cancer [76]. All these findings suggest that the metabolic aberrations that contribute to sustaining the S1P level, even if different in different tumors, are functional to cancer malignancy, and independent of individual genetic fingerprints, which vary greatly in different cancers as well as in individual cancer types.

Concerning brain tumors, it is surprising that, despite the implication of S1P in cancer development and malignancy, no studies to date have reported on S1P levels, metabolism, and signaling in human brain cancers other than gliomas.

## 5. The Tumor Microenvironments and the Specific Features of the GBM Ones

Cancers are not just as an assemblage of malignant cells, but rather develop as aberrant organs, characterized by intricate components, including various types of non-tumoral host cells and their extracellular environments [77]. The tumor microenvironment (TME) plays a critical role in tumor development, and acts as a specialized array of niches with typical traits, where the cross-talk between the tumor cells and the TME components acts as a key player affecting malignant properties and tumor progression [78]. Cell-to-cell interactions emerged as crucial contributor to this process, which occurs via soluble molecular signals by the hypoxic environment of the growing tumor. The TME influence on cancer progression is a dynamic process, undergoing changes in response to varying environmental conditions and extracellular signals, along with cancer progression.

GBMs display a high degree of intratumor heterogeneity, and the TME markedly contributes to this diversity, underscoring the need to understand how GBM cells drive the construction of their own TME. Increasing evidence supports the GBM microenvironment as an important, complex component in GBM, which involves communication with, and manipulation of, other brain cells, and tremendously influences the tumor progression, spread, and treatment resistance [77]. Indeed, GBM tumor cells can be exposed to diverse microenvironments, consisting of an array of non-neoplastic cell types, comprising of resident and infiltrating immune cells, brain endothelial cells (bECs), neurons, astrocytes, and fibroblasts. These different cell populations can secrete soluble signals, cytokines and growth factors and extracellular matrix (ECM) components, which enrich the GBM microenvironment with specific repertoires of signal/regulatory molecules [11,79,80,81]. These environmental cues create an inflammatory environment, and steer GBM cell fate through influencing their quiescence, proliferation, survival, stemness, and invasion, modulating the biological functions of the infiltrating cells to further support the growth of cancer cells, their invasion, and resistance to therapy.

Of relevance, the TME plays a key role in GSC properties too, leading to an enrichment of their malignant potential [82]. In the central part of GBM, necrotic areas are present, surrounded by “pseudopalisading” tumor cells that deal with hypoxia and nutrient starvation, and activate migration processes in an attempt to escape hypoxia and reach well-vascularized and oxygenated areas [83]. Besides oxygen and nutrient scarcity, these hypoxic niches are highly acidic, thus selecting GBM cells that are able to withstand hard conditions. Among them, some GSCs can survive, living as quiescent cells in hypoxic areas. In GSCs, hypoxia inducible factor (HIF) activation occurs, leading to the expression of metabolic enzymes, ABCTs, and growth factors, and thus potentiating angiogenesis, immune suppression, and therapy resistance [84,85,86,87]. GSCs also reside in perivascular niches, which are proximal to the invasive tumor edge and close to the disorganized and leaky tumor blood vessels [88], and an intensive cross-talk between them and ECs promotes angiogenesis, GSC self-renewal, migration, and survival [89,90].

## 6. The Cellular Contributors to S1P in the Tumor Microenvironment

In the complex array of signals participating in the TME, accumulating evidence indicates that S1P represents a key component with multiple cells including tumor ones and the tumor-recruited brain normal cells, contributing to its enrichment in the GBM microenvironment (Figure 2).

In order to make possible S1P secretion and its extracellular cross-talk with different cells in the GBM microenvironments, some important mechanisms of transport and/or communication among those cells likely play an utmost role. Some ABCTs, including ABCA1, ABCA7, ABCB1 (or P-glycoprotein or MDR1), ABCC1 (MDR-associated protein-1), ABCC4 (MDR-associated protein-4), and ABCG2 (breast cancer resistance protein), as well as Spns2 and Mfsd2b, are the known transporters that transfer S1P through the cell plasma membrane [33,34,36,91,92]. Surprisingly, the role of these transporters in the S1P secretion by GBM and in the S1P interactions between GBM and non-tumoral cells in the TMEs remains poorly investigated.

In the following sessions, we review the current knowledge on S1P contribution of the different tumoral and non-tumoral cells involved in the export of S1P, and thus in its enrichment in the GBM microenvironments.

### 6.1. S1P Secretion by GBM Cells and GSCs

Once produced inside GBM cells, S1P can be exported into the extracellular milieu, in an “inside-out” signaling event crucial for the autocrine and paracrine first messenger action of S1P through its specific receptors. Edsall et al. [93] initially provided evidence on the extracellular release of S1P by rat C6 glioma cells, but not by PC12 pheochromocytoma cells, suggesting that not all tumor cells are equipped with the molecular mechanisms subtending S1P export. Further studies reported that different human GBM cell lines, including U87MG, CCF-STTG, and T98G, can all constitutively secrete S1P [50,94,95,96], underlying the relevance of GBM cells in the secretion of S1P in the extracellular TME.

The treatment of U87MG cells with the SphK1 inhibitor SKI-1a markedly reduced extracellular S1P, supporting the role of SphK1 in providing S1P for export [50]. Importantly, different signaling mediators strictly connected to GBM malignancy can increase the basal expression and activity of SphK1. These stimuli include basic fibroblast growth factor (bFGF), epidermal growth factor (EGF), vascular endothelial growth factor (VEGF), tumor necrosis factor α (TNF-α), platelet-derived growth factor (PDGF), insulin-like growth factor binding protein-3 (IGFBP3) [10,97,98,99,100,101,102,103], and, not least, S1P itself [104,105]. These molecules induce a rapid and transient activation of SphK1, its subsequent translocation to the plasma membrane, and export of newly synthesized S1P, and can exert long-term effects by enhancing the SphK1 expression [106,107]. More than 30 years ago, Libermann et al. reported that the EGF receptor is overexpressed and constitutively activated in GBM [108]. Nowadays, this feature is considered an important event in the pathogenesis of a subset of GBMs. EGF stimulates SphK1 activity and induces its translocation from the cytosol to the plasma membrane through sequential activation of c-Src and PKCδ [99,109]. Moreover, it was found that interleukin 1 (IL-1), a pro-inflammatory cytokine secreted by GBM cells in relevant amounts [110], upregulates SphK1 at the transcription level, resulting in a correlation between its level and that of SphK1 in GBM cells [111]. In human GBM cells, IL-1 induces transcription of SphK1 by a novel activating protein 1 element located within the first intron of the SphK1 gene, which can be blocked by inhibition of JNK [111]. Hypoxic stress also increases the expression of SphK1 in GBM cells, through an increase in HIF-2α activity that binds to the promoter of the SphK1 gene, leading to increases in not only intracellular but also extracellular S1P [112,113]. Of interest, the SphK1/S1P-signaling pathway emerged as a regulator of HIF-2α expression in multiple cancer cells, including GBM ones [114], suggesting a positive feedback loop between SphK1/S1P and HIFs in response to hypoxia. Similar to the HIFs, the LIM-only transcription factor LMO2, an important regulator of angiogenesis that promotes angiogenic traits in GSCs [115], binds to the SphK1 gene and increases the expression of SphK1 protein [116].

Despite these numerous reports underlying the role of SphK1 in S1P export by GBM cells, very recently Neubauer et al. [117] demonstrated that SphK2 plays a role too. In particular, it was found that SphK2 interacts with the cytoplasmic DYNC1I1-containing dynein complex, and this interaction appears to facilitate transport of SphK2 away from the cell periphery. In GBM cells, a dramatic downregulation of DYNC1I1 correlated with poorer patient survival, and paralleled with SphK2 localization to the plasma membrane and formation of extracellular S1P [117],

It should be added that both SphK1 and SphK2 are secreted by some normal and tumor cells as a component of shed vesicles [23,118,119,120], likely indicating that SphKs localized in these vesicles cause sustained extracellular S1P production. However, the role of vesicle-mediated formation of extracellular S1P in the GBM microenvironment remains largely unknown.

In the U87MG cell line, the ABCA1 transporter, which facilitates S1P efflux in astrocytes, was found involved in the S1P efflux, thereby contributing to inside-out signaling [121]. ABCA1 expression in this GBM cell line is potently induced by 25-OH-cholesterol [121], a natural oxysterol synthesized and secreted by GBM cells [122], suggesting that oxysterol synthesis provides an autocrine signal that enhances ABCA1 expression and S1P export from GBM cells.

Further relevant findings refer to S1P metabolism in GSCs, the small population of GBM cells crucial for its malignancy. We recently reported that GSCs are equipped with an efficient molecular machinery that allows them not only to rapidly form S1P, but also to efficiently export it [95,123]. Importantly, GSCs recently emerged as the cells with the greatest synthesis and secretion of S1P in GBM, the amount of extracellular S1P provided by GSCs accounting for about one order of magnitude higher than that by GBM cells [95]. It is worth noting that, despite similarities between neural stem cells and GSCs [124], GSCs appear to possess the unique ability to export S1P and enrich their extracellular milieu with S1P, as neural stem cells are incapable of this export [125]. The efficiency of S1P export by GSCs is strictly dependent on the availability of its substrate sphingosine [123], which can be released from necrotic cells. The enrichment of GSCs in the perinecrotic niche of GBM [126] suggests that S1P biosynthesis and release occur very rapidly and to a high extent in this niche, providing these cells with a favorable microenvironment. Importantly, the proliferative and stemness properties of GSCs relate to a gain in extracellular S1P, paralleled by a downregulation of intracellular ceramide levels due to increased metabolic flux to complex sphingolipids [9,123]. Thus, an imbalance of the sphingolipid rheostat, with the S1P domination in the GSC niches, occurs, and provides an advantage to the malignant qualities of GSCs. Moreover, the investigation of the effect of EGF and bFGF, recognized autocrine signals in GSCs, revealed that the constitutive S1P secretion by GSCs is proliferation-dependent, and significantly enhanced by the presence of these growth factors [123]. Up to now, the mechanism underlying S1P export by GSCs is unknown. It should be noted that different ABCTs, known as transporters of S1P through the cell plasma membrane, and particularly ABCB1, ABCC1, and ABCG2, are highly expressed in GBM cells, and particularly in GSCs [127,128,129]. Their expression is correlated with GBM aggressiveness [130], suggesting their possible involvement in the S1P export from GSCs/GBM cells.

Despite the influence of GBM interactions with GSCs emerging [131,132], up to now it is unclear whether GBM cells contribute to S1P secretion by GSCs and/or vice versa.

### 6.2. S1P Secretion by Non-Cancer Cells Recruited in the GBM Microenvironment

The TMEs in which GBM cells develop and grow largely contribute to its heterogeneity. GBM microenvironments contain an array of non-neoplastic cells, including infiltrating and resident immune cells, vascular cells, and other glial cells. GBM cells subvert normal brain cells to create microenvironments that contribute to crucial GBM properties and favor tumor success. Indeed, GBM cells recruit different normal brain cells, such as astrocytes, innate immune cells, and ECs, and change their phenotype, inducing them to modify the GBM microenvironments with pro-tumoral signals [133,134,135,136]. In the following sections we overview the cross-talk between GBM cells and normal brain cells, and focusing on the contribution of non-tumoral cells to the S1P level in the GBM microenvironments.

#### 6.2.1. Microglia and Macrophages

Two types of immune cells, called tumor-associated macrophages (TAMs), including intrinsic microglia, and peripheral-recruited macrophages, represent the majority of the non-neoplastic cells in GBM [137]. Microglial cells are recruited at the tumor site by GBM-secreted chemoattractant factors [138,139], while blood-derived macrophages accumulate in GBM through the breakdown of the blood–brain barrier (BBB) [140]. Both these TAMs can release a wide array of growth factors and cytokines in response to factors produced by GBM cells, which create a supportive stroma for neoplastic cell expansion and invasion, and facilitate GBM progression [88,141].

Different investigations performed on microglial cells demonstrated an important role of S1P in the pathogenesis of inflammation [142,143,144,145], including neuroinflammation [146]. Both macrophages and microglial cells are able to secrete S1P extracellularly; inflammatory stimuli, such as lipopolysaccharide (LPS), upregulate SphK1 expression on the plasma membrane in these cells and significantly increase S1P secretion from microglia [147,148,149]. Microglia/macrophages express different ABCTs shown to mediate the S1P efflux, including ABCB1, ABCC1, and ABCG2 [150,151,152], and their expression changes after LPS activation [150]. In addition, a very recent report showed that the S1P transporter Spns2 is expressed and functions as a transporter of S1P export from microglia, promoting microglia pro-inflammatory activation in vitro and in vivo [153].

Despite the current knowledge on S1P from TAMs in GBM being limited, TAMs are among the cell types that can contribute to enriching the GBM microenvironment with S1P, and to promote GBM progression (see below).

#### 6.2.2. Endothelial Cells

GBM has highly abnormal and exuberant angiogenesis, the formation of aberrant tumor vasculature representing a key event in its growth and malignancy [76]. Indeed, endothelial activation significantly contributes to the GBM microenvironment and acts as a critical regulator of GBM progression [77].

Interestingly, SphK1 is overexpressed in ECs from brain tumors, and S1P stimulates ABCB1 expression and transport, suggesting that SphK1 and S1P could contribute to the multidrug resistance phenotype in brain tumor-derived ECs [154]. Very recently, we reported that bEC lines, as well as primary bECs derived from different GBM patients, exhibit the capacity of spontaneously secreting S1P into the extracellular environment in significant amounts [96]. A small fraction of S1P is produced in the bEC-conditioned medium [96], suggesting that S1P may be also produced extracellularly by an extracellular SphK, as it occurs in human umbilical vein ECs [23]. Most importantly, co-culture with human GBM cells induces overexpression, plasma membrane translocation, and enhanced activity of SphK2, which leads to a potent stimulation of S1P export by different bECs [96]. Therefore, GBM-EC co-culture induces an enhancement of S1P synthesis and export by bECs, this last possibly through ABCB1 or Spns2 [155]. These findings indicate that bECs are able to increase the rate of S1P production and release in response to GBM co-culture, suggesting they stimulate GBM cells to produce diffusible factor(s) that favors their “inside-out” S1P signaling in the absence of cell-to-cell contact.

#### 6.2.3. Neurons and Astrocytes

Previous studies demonstrated that primary cultures of cerebellar and hippocampal neurons express SphK1 and have the capacity to secrete S1P in the extracellular milieu [156,157]. This secretion is potentiated by neuronal depolarization, PKC activation, and by the glutamate neurotransmitter. Upon stimulation, SphK1 was shown to translocate to the plasma membrane and produce S1P for autocrine S1P receptor signaling [156,157]. It was reported that exogenous S1P elicits glutamate secretion, and potentiated depolarization-evoked secretion from neurons [156], suggesting that a depolarization/S1P secretion cycle occurs. Despite the detrimental effects of GBM progression on neuron survival and function [158,159], the influence of GBM on S1P production by the neuronal population remains unknown. Interestingly, GBM has the ability to release glutamate in the nearby environment [160], and a clinical study involving GBM patients found peritumoral glutamate levels 100-fold higher than those in an uninvolved brain [161]. Thus, it appears likely that neuronal proximity to GBM cells may favor S1P enrichment in GBM microenvironments.

Different is the knowledge about astrocytes, which emerged as important cellular sources of S1P in the central nervous system (CNS), and which are potentially involved in the GBM microenvironments. Indeed, these cells are able to efficiently export S1P [156], and bFGF, a crucial factor in GBM progression and stemness, as well as in GBM resistance to anti-VEGF therapy [162,163,164], was found to act as a potent stimulus of S1P export [156]. Of note, the ABCA1 inhibitor glyburide was reported to induce an accumulation of intracellular S1P and exogenously added S1P partially restored astrocyte proliferation in the presence of this inhibitor [156]. These and other findings [165] appear to support a critical role for ABCA1 in mediating this export. Moreover, very recently, Dréan et al. [130] showed that ABCG2 is highly expressed by astrocytes, suggesting its possible involvement in S1P efflux from these cells.

## 7. S1P Role in Cancer Hallmarks

As a malignant tumor, GBM exhibits hallmark capabilities that are crucial to cancer phenotypes [166]. Hallmark characteristics of GBM include uncontrolled proliferation, invasiveness, stemness, intense angiogenesis, and death resistance, and account for GBM’s resistance toward radiotherapy and chemotherapy and poor prognosis. Very recently, enabling replicative immortality, inducing angiogenesis, reprogramming cellular energetics, and evading immune destruction emerged as the challenge to find similarities in GBMs and as the most promising GBM hallmarks for clinical impact [167]. Of relevance, S1P in the GBM microenvironments has been implicated in regulating key properties underlying GBM malignancy and deadly features [9,10], participating in the different mechanisms known to sustain the hallmarks of cancer (Figure 3).

Despite the suggestion that S1P may function as an intracellular signal [4,168], most of its actions in cancer cells occur through its acting as a high-affinity agonist at its five known G protein-coupled receptors, and downstream activation of their signaling pathways [169]. S1P receptors are a family of G protein-coupled receptors belonging to the endothelial differentiation gene (EDG) receptor family, which are involved in the signaling of the lysophospholipids S1P and lysophosphatidic acid (LPA) [170]. The S1P1–5 members of this family, which are specific for S1P, are able to induce multiple biological pathways strictly related to cancer biology [169]. It was shown that different human GBM cell lines and GBM specimens express the genes of S1P1–3 and S1P5 receptors. Increased levels of S1P1, S1P2, and S1P3 were observed in GBM tissue specimens, and signaling of S1P1 and S1P2 were markedly correlated with patient survival rates [57]. Despite Yoshida et al. [58,171] reporting that S1P1 expression is downregulated in GBM specimens, which enhances tumor cell proliferation and correlates with shorter survival of GBM patients, recent investigations demonstrated an inverse correlation between S1P1 expression and GBM patient survival [50,56,57]. Whether these opposite findings depend on the heterogeneity of GBM or experimental conditions remains to be clarified.

S1P1–3 are linked to several, often interconnected signaling pathways, including mitogen-activated protein kinase (MAPK), c-Jun N-terminal kinase (JNK), extracellular signal-regulated (ERK/MAP) kinase, phosphatidylinositol-4,5-bisphosphate 3-kinase (PI3K), phospholipase C (PLC), phospholipase D, and other downstream mediators [1,22,168]. S1P can stimulate diverse signal transduction pathways in different cell types, as well as within the same cell, depending on the expressed S1P receptor pattern, as well as their differential coupling to different G proteins. S1P1 is known to be coupled exclusively to G_i_ protein, leading to cAMP reduction and activation of Ras, MAPK, PI3K, Akt, and PLC pathways. Both S1P2 and S1P3 are coupled to G_i_ and G_12/13_ proteins, and activate Ras, MAPK, PI3K, serine-threonine kinase Akt, PLC, and Rho-dependent pathways. G protein-coupling properties of S1P4 and S1P5 are less clear at present. They couple to G_i_ and G_12/13_, to activate Rho-dependent pathways, and mainly influence cell motility and migration.

In addition to transducing S1P signaling, the G protein-coupled S1P receptors have been implicated in signal amplification of a variety of growth factor receptors, via receptor transactivation [172]. Moreover, transactivation of S1P receptors by growth factors may also occur [173,174]. This transactivating cross-talk likely results in the potentiation of downstream signaling pathways, and thus of cell malignancy.

Very recently, El Buri et al. [175] have identified a new novel mechanism by which S1P regulates the TME. The authors found that, like the SphKs, the S1P2 receptor was released through exosomes into the conditioned medium from a breast cancer cell line, and induced robust ERK1/2 activation and proliferation of fibroblasts. Despite this interesting finding, to date it remains unknown if the release of exosomal S1P receptors occurs in GBM and/or other cancers, and contributes to their properties.

The features of S1P-signaling pathways in cancer, including GBM, are complex and interconnected and, through autocrine and paracrine signaling as well as overexpression of S1P receptors, lead to high cell responsiveness. In the following part of this paper, we describe the actual knowledge on the S1P signaling that pertains to their specific involvement in GBM hallmarks.

### 7.1. S1P in the Cancer Microenvironment Promotes Sustained Proliferation

Among the different properties of GBM, there is a highly proliferative potential, and the tumor microenvironment is essential to control proliferation. Of relevance, excessive growth factor signaling is a crucial component of GBM malignancy, including the abundance of EGF, bFGF, and VEGF [80]. The signaling of all these factors involves S1P as mediator, as they induce a rapid and transient activation of SphK1, its translocation to the plasma membrane and S1P export, and can exert long-term effects by enhancing SphK expression. In particular, several reports demonstrated that SphK1 has an important growth-regulatory role in different cancer cells, and plays a crucial role in the mitogenic action of S1P in human GBM cells, too. Indeed, SphK1 inhibition results in growth arrest of GBM cells both in vitro and in vivo [54,55,176,177]. The highly proliferative areas of GBM are close to the central necrosis, where hypoxia is at its highest expression [178], and this environment promotes SphK1 and S1P export (see above). Among different growth factors, IGFBP3 is overexpressed in GBM [102], and its expression is associated with poor patient outcome [179]. IGFBP3 functions as a potent regulator of GBM cell proliferation, and its action as a tumor promoter occurs through increasing SphK expression and S1P formation [180]. Notably, S1P has been reported to stimulate the production and secretion of EGF, PDGF and VEGF [181], as well as to transactivate EGF/IGF receptor-signaling pathways, leading to enhanced GBM cell proliferation and tumor growth [172]. Thus, S1P-induced transactivation of these receptors appears to constitute an efficient switch to integrate rapid G protein-dependent signals into long-term cell proliferation.

SphK1 was also found to be involved in the proliferative effect of IL-1, which upregulates this kinase [111]. Moreover, SphK2 knockdown was found to inhibit GBM cell growth more potently than SphK1 knockdown [54], suggesting that both SphKs can participate to promote GBM cell growth. Consistently, both SphK1 and SphK2 isoforms were found involved in the proliferation of GBM cells in hypoxic conditions [113]. The proliferative effect of the activation was reported to occur through ERK-dependent activation for SphK1 and ERK-independent activation for SphK2, and the cell cycle was arrested in G2⁄M phase or in S phase, respectively [113].

The proliferative role of extracellular S1P in GBM is further substantiated by several reports demonstrating that human GBM cells respond mitogenically to nanomolar concentrations of S1P [57,96,121,182], and that extracellular S1P antagonized the inhibition of cell proliferation induced by inhibition of the S1P transporter ABCA1 [121]. The proliferating effect of S1P was found to involve its binding to, and activation of, different S1P receptors, including S1P1–3 and S1P5, which are expressed in different GBM cell lines and human GBM specimens [54,57,58,96,171,182]. The S1P binding to all these receptors is able to affect GBM proliferation due to the ability of these receptors to activate G_i_ (all S1P receptors) and/or G_12/13_ (S1P2, S1P3 and S1P5) [121]. The activated signaling pathways lead to the activation of ERK/MAP kinase, through both PI3K-dependent and independent pathways [182,183]. Through the stimulation of ERK/MAP kinase, S1P1 induces the expression of bFGF [184], and it was shown to exert a potent effect on GBM cell proliferation due to its high potency for ERK activation [185]. In agreement, SphK1 was shown to contribute to EGF-induced GBM cell growth by enhancing activation of ERK signaling [186]. Interestingly, we recently reported that brain ECs and GBM-derived ECs co-cultured with GBM induce increased expression of S1P1 and S1P3 in GBM cells, which promote S1P-induced proliferative properties in GBM cells [96].

It was shown that the overexpression of S1P5, opposite to that of S1P1–3, decreases cell proliferation in GBM cells [56], suggesting that a balance among different S1P receptors participates in the control of GBM proliferation.

Notably, the capacity for extensive proliferation and self-renewing is crucial not only in GBM cells, but is also a key determinant of GSCs [187]. Using GSC lines derived from human GBM specimens with different proliferative indexes, we found that S1P promotes GSC proliferation, cell cycle progression, and stemness phenotypic profile [123]. In addition, the export of S1P was enhanced in the cells exhibiting a high proliferative index, suggesting that S1P may act as an autocrine signal to maintain a pro-stemness environment, and favoring GSC proliferation and stem properties [123]. Interestingly, besides inducing the growth pattern of GSCs, S1P induced different stem cell markers in GSCs, thus enhancing their stemness phenotype [123]. Since S1P was unable to induce the expression of stemness markers in non-stem primary GBM cells, it appears that the S1P-promoting stemness resides in its ability to favor the selective expansion of GSCs.

The signal transducer and activator of transcription 3 (Stat3) is constitutively activated by phosphorylation in GSCs, promotes GSC proliferation, and is indispensable for GSC-induced tumor formation [188]. Of relevance, Stat3 has S1P1 as a transcriptional target, and, in turn, S1P1 signaling is required for persistent activation of Stat3, thus creating a positive feedback loop that fuels the growth of a range of cancers, including GBM [189]. Hirata et al. [190] found that in GSCs and in several types of cancer stem cells, activation of G_i_-coupled S1P3 by S1P results in the induction of proliferation through binding and activation of Notch signaling, a key stem cell pathway. Moreover, a very recent study reported that, in both GBM cells and GSCs, the response to S1P stimulation through G_12/13_ coupled receptors involves the activation of RhoA, which results in divergent signaling pathways, both required for the nuclear accumulation of the transcriptional coactivators MRTF-A and YAP, as well as for cell proliferation [191]. S1P stimulates cell proliferation only when both MRTF-A and YAP can be activated [191], implicating that more than a single signaling pathway needs to be turned on to increase the rate of GBM proliferation.

Overall, S1P promotion of GBM cell and GSC proliferation appears to occur as a complex, dysregulated and fail-safe mechanism, implicating the involvement of different receptors and multiple signaling pathways, including divergent molecular interactions (Figure 4).

### 7.2. S1P in the Cancer Microenvironment Promotes Invasive Behaviour

GBM is an extremely aggressive tumor which exhibits an exceptional ability to invade the surrounding parenchyma, and widely infiltrates normal brain, typically disseminating along blood vessels or nerve fibers, and reaching brain regions many centimeters away from the main tumor mass [39,192]. The highly infiltrative nature of GBM makes complete surgical resection with clean margins nearly impossible, and gives rise to recurrence, a process that occurs in almost all GBM patients despite aggressive surgical resection and chemotherapy [39]. Moreover, although metastatic GBM is rarely observed, GBM can spread hematogenously [193], and numerous cases of extracranial metastasis in GBM patients are reported [194]. In addition, there is accumulating evidence that current therapeutic modalities, including anti-angiogenic therapy and radiotherapy, can enhance glioma invasiveness.

GBM cells secrete several factors that promote their motility through autocrine/paracrine signaling [195]. Different studies have implicated SphK/S1P signaling in enhancing cancer cell migration, invasion and metastasis, the effects of S1P varying between different cancers and even among different cells of the same cancer [196]. Studies on cell migration demonstrated that different S1P receptors exert opposite effects on GBM cells. Indeed, S1P1 and S1P3 signal migratory responses and amplify those exerted by other growth factors, whereas S1P2 signals inhibition of growth factor-evoked migration [185,197]. The possible reason of this antagonistic effect appears to reside in the fact that S1P1 and S1P3 stimulate _the_ small GTPase Rac, whereas S1P2 inhibits it [198]. Since the S1P2 activation of G_i_ is by far less efficient than that of S1P1 and S1P3, it was hypothesized that the robust activation of the G_12/13_-Rho pathway by S1P2 likely masks its G_i_-mediated Rac stimulation, resulting in inhibition of cell migration [199].

Opposite to the negative effect of S1P2 on cell migration, the activation of all S1P1–3 receptors results in a potent increase in GBM cell motility and invasiveness of different GBM cell lines. In particular, S1P emerged as a key signal in the regulation of GBM invasion, being able to stimulate different events involved in this complex process, including reduction of cells and of extracellular matrix (ECM) adhesion, cytoskeletal remodeling, matrix metalloprotease (MMP) secretion, and then ECM degradation (Figure 5).

Overall, G protein signaling of the S1P1-3 subtypes results in: (1) Rac activation [198], which regulates cell morphology and actin dynamics, and stimulates cell squeezing through the narrow extracellular spaces that are typical of the brain parenchyma; (2) increased secretion of the matricellular protein CCN1/Cyr61 that, once secreted, binds to integrin αV-β3 to enhance ECM adhesion [185,200,201]; (3) activation of the signaling cascades MEK1/2, PI3-kinase/AKT1 and Rho-kinase, which leads to enhanced gene and protein expression of the plasminogen activator system proteins, including the urokinase-type plasminogen activator (uPA), plasminogen activator inhibitor 1 (PAI-1), and a receptor for uPA (uPAR) [57,183,202].

As for the potent induction of GBM cell invasiveness by S1P, the upregulation of the pro-invasive molecule CCN1, neurogulin-1 (NRG-1), uPA and its receptor has been shown to be involved [185,200,203]. Matriptase, upregulated through S1P-receptor signaling, is also secreted, and activates uPA that induces MMP activation, leading to the activation of plasminogen to plasmin. On its turn, plasmin, as such, or by activating MMPs, promotes the degradation of extracellular matrix and cellular invasiveness (Figure 5).

Another very important signal in the migration and invasion of GBM cells by S1P is Ca^2+^ mobilization [204,205], a signaling event linked to most S1P receptors [206,207]. The Ca^2+^ gradient in GBM cells allows them to move in a specific direction, and activates calpain 2, a cysteine protease important for the invasion of GBM cells within the dense microenvironment of the brain [208]. S1P5 was found as responsible for stimulation of PLC-Ca^2+^ system in C6 glioma cells [184], and S1P2 later appeared as the most likely factor responsible for PLC activation and Ca^2+^ release in GBM cells [197]. Of interest, the competitive inhibition of the microsomal glucose-6-phosphate transporter (G6PT), which can enhance Ca^2+^ sequestration in the ER [209], was shown causing a downregulation of S1P-induced cell migration [210]. Moreover, it was found that gene silencing of not only the G6T but also the membrane-type 1 MMP (MT1-MMP) decreased the extent of S1P-induced Ca^2+^ mobilization, unrevealing these signaling pathways that are required in GBM cells for efficient Ca^2+^ mobilization, and invasive effects in response to S1P [204].

It is important to highlight that different signaling pathways can influence the S1P-dependent GBM cell migration and invasion. For instance, EGF and IL-1 signaling pathways may enhance S1P-dependent expression of PAI-1 and uPAR [99,201]. In addition, PDGF-induced GBM chemotaxis is dependent on S1P, which in turn activates the transient receptor potential channel TRPC1, leading to the entry of Ca^2+^ and increased cell invasivity [211].

Finally, it emerged that S1P is relevant also in the potent invasive properties of GSCs. In particular, a population of GSCs isolated from the U87MG cell line exhibited an increased migratory response to S1P compared with parental cells, and combined regulation of S1P/LPA-mediated signaling and MT1-MMP are involved in the invasive properties of GSCs [212]. In addition, the high invasive potential of GSCs was found correlated with the high expression of S1P1 [200,212].

### 7.3. S1P in the Cancer Microenvironment Promotes Death Resistance

In different cancer cells, S1P acts as a key actor both in promoting survival pathways and in antagonizing the players of death signaling, including those involving ceramide. It is widely recognized that the balance between ceramide and S1P levels in GBM cells plays a crucial role in determining cellular fate, and leads to cell death when ceramide prevails or to cell survival when S1P levels are increased. The dysregulation of the ceramide/S1P rheostat appears crucial in the death-resistance features of GBM [9]. There is strong evidence demonstrating the role of ceramide as a death-inducing signal, involved in the toxic effects of chemotherapeutic drugs and radiations, as well as that of S1P in promoting cell survival in different cancer cells [6,213]. To antagonize autophagic/apoptotic death induced by chemotherapeutic drugs, chemotherapy-resistant GBM cells are able to maintain reduced levels of ceramide [214]. In addition, the upregulation of S1P represents a key strategy to survive and to enhance their death resistance properties of GBM cells, GSCs and human GBM tumors [9]. Indeed, GBM cells and GSCs use S1P to maintain and promote their survival, even when submitted to toxic treatments, such as radio-chemotherapy.

S1P plays a very important role in apoptosis, and the activity of SphK1 was found directly related to the survival of cancer cells and acquisition of replicative immortality, in a process termed ‘non-oncogenic’ addiction [215]. Several studies support a key role of SphKs in the promotion of S1P-induced survival in GBM, and SphK inhibition/genetic ablation sensitizes GBM cells to chemotherapeutics, and slows GBM growth in mice. A pivotal study reported that SphK1 expression correlates with poor survival of patients with GBM [54], and the following investigations revealed the importance of this enzyme. In particular, it was found that SphK1 inhibition: (1) leads to cell death by inducing apoptosis of human GBM cells and xenografts, and reduces survival in orthotopic GBM [72,177]; (2) sensitizes GBM cells, and different cancer cells, to several cytotoxic drugs [216]; (3) is effective in potentiating the cytotoxicity of both TMZ and radiation therapy in various human GBM cell lines [217,218]; and (4) induces apoptosis and inhibits colony formation in TMZ-resistant GBM cells [219]. Of relevance, recent studies on GBM irradiated cells revealed that S1P is upregulated, and SphK1 gene significantly induced following radiation, suggesting that S1P links to radio-resistance and increased aggressiveness of irradiated GBM cells [69,220].

The mechanisms underlying the pro-survival properties of S1P mainly include signaling pathways that result in the inhibition of apoptosis and/or the induction of protective autophagy [9]. S1P-mediated protection from apoptosis appears to occur primarily through G_i_-mediated activation of PI3K/Akt/eNOS signalling, and ERK and p38MAPK have been implicated too [221]. S1P-mediated inhibition of apoptosis further involves inhibition of cytochrome c release, activation of caspases, and activation of the stress-activated protein kinase JNK.

The relevant role of S1P in GBM survival is strengthened by accumulating reports on S1P antagonists as potential targets for cancer therapies. The survival effect of S1P in GBM cells has been related to the activation of different survival pathways, and among them the PI3K/Akt signaling pathway that is involved in the pathogenesis of GBM and plays a crucial role in conferring GBM resistance to cytotoxic treatments [222,223,224]. S1P can induce robust Akt activation, most probably through a Gαi signaling pathway, which in turn signals to a variety of key downstream molecules, finally suppressing cell death and promoting cell survival [225]. Indeed, targeting SphK1 in GBM cells with SK1-I rapidly reduced Akt phosphorylation, inhibited JNK, and finally reduced death in GBM cells in vivo [177]. Through the overexpression or downregulation of SphK1 in GBM cells, it was shown that S1P suppresses apoptosis induced by radiation and chemotherapeutic drugs via Akt activation, subsequent inactivation of FOXO3a and of Bim, and finally downregulation of the proapoptotic Bcl-2-like protein 11 (Bim) [226]. Additionally, and consistent with the survival role of S1P, the S1P receptor antagonist FTY720 induces apoptosis in GBM cells [227,228].

Opposite to the S1P-mediated activation of Akt, ceramide acts as a potent inhibitor of this signaling pathway. Notably, Akt stimulation in GBM cells also promotes the ER–Golgi trafficking of ceramide and its use for the synthesis of complex sphingolipids, leading to a reduction of ceramide levels and inhibition of ceramide-induced apoptotic and non-apoptotic cell death [229]. Thus, the S1P-promoted Akt activation results not only in the promotion of survival signaling pathways, but also in the inhibition of ceramide-induced cytotoxicity. It is worth noting that that the bioflavonoid luteolin, recently proposed as an effective anticancer agent for different human neoplasia including GBM [230], exerts a potent cytotoxic effect by increasing ceramide and downregulating S1P and Akt activation [231]. On this basis, it is likely that the high levels of S1P in human GBM might contribute to the intrinsic resistance of GBM to cytotoxic treatments also by inducing low ceramide levels. In support, ceramide levels in GBM samples inversely relate to tumor progression and survival of GBM patients [49,50].

Currently, it is assumed that GBM recurrence results from GSCs, which are intrinsically radio- and chemo-resistant, highly invasive, enriched after radiotherapy, and directly associated with poor patient prognosis [136]. Different reports found that S1P plays a key role in GSCs resistance to TMZ [95]. Indeed, the inhibition of S1P biosynthesis made GSCs sensitive to TMZ, and exogenous S1P, alone or in combination with TMZ, was able to revert the cytotoxic effect of TMZ–SphK inhibitor co-treatment, promoting GSC survival. Annabi et al. [212] reported that GSCs are much more responsive to extracellular S1P than their parental GBM cells, with differences in S1P-receptor expression contributing to this feature. Since S1P1 expression is increased in GSCs from U87MG GBM cells, and promotes cell survival in cultured cells and in a murine model of intracranial GBM [212], it appears reasonable that S1P1 might be involved in the pro-survival effect of extracellular S1P in TMZ-treated GSCs. In support, it was reported that administration of the functional S1P antagonist FTY720 to nude mice led to downregulation of S1P receptors, induced apoptosis in GSCs, and was synergistic with TMZ in promoting cytotoxicity [232]. Of relevance, the apoptosis induced by inhibiting SphKs was shown to be highly effective, and to specifically target GSCs [220], known as resistant cells to the standard GBM chemotherapy agent TMZ.

### 7.4. S1P in the Cancer Microenvironment Promotes Immune-Evasion

Among the different non-neoplastic cells of the GBM microenvironments, cells of the immune system, and especially TAMs, play a critical role in the regulation of tumor malignancy [233]. These cells are the dominant non-tumoral population, accounting for 30%–40% of the cells in the tumor, are inversely correlated with patient survival [234,235], and engage in reciprocal interconnections with tumor cells to promote GBM growth, progression, and invasion [236]. Consistently, in both animal xenografts and surgical resections, the invasive front of GBM, where GSCs reside, contains abundantly infiltrating TAMs [237], and in in vivo GBM models, depletion of microglia/macrophages significantly reduces tumor growth [238].

Microglia and macrophages can possess highly diverse phenotypic and functional heterogeneity by polarizing to either M1 or M2 sub-types, which are representative of anti- and pro-inflammatory phenotypes. It is believed that tumor-associated macrophages are mainly M2 macrophages, and more likely contribute to tumor growth, rather than exerting effective antitumor protection [239], and M2 abundance is associated with poor prognosis for patients with different tumors including GBM [240,241]. Microglia/macrophages within GBM appear to initially participate in tumor surveillance as M1 type, but are then subverted by GBM to adopt M2 anti-inflammatory phenotypes, and subsequently promote immunosuppression, tumor angiogenesis, and invasion [242]. TAMs can also secrete immunosuppressive factors, such as IL-4, IL-10, and TGF-β1, which in turn polarize M1 cells into the M2 type, which suppresses antitumor-immune responses [243]. In further support, a recent immunogenomic analysis of 33 distinct cancer types classified GBM among tumors with a greater range in leukocyte fraction compared to other cancer types, with a prominent M2-like macrophage signature, and a high anti-inflammatory macrophage response [244].

Several studies have identified an essential role for S1P and its receptors in immune responses [143,146,245]. Microglial cells and macrophages express all five S1P receptors (S1P1–5), and their expression varies according to the activation state of these cells [148,246,247]. S1P1 receptors, highly expressed on naïve macrophages, are decreased in both M1- and M2-polarized cells, while S1P4 is reduced only in M1-polarized cells [247]. S1P induces monocyte migration in a receptor-specific fashion, with S1P1 and S1P3 receptors being involved in the migration of pro-inflammatory M1 and anti-inflammatory M2, respectively [248,249,250]. Because the migration potential of M1 is higher than that of M2 macrophages, it has been suggested that the ratio between S1P1/S1P4 receptors orchestrates macrophage migration [247]. The elevated S1P content within GBM tissue [50] and the reduced circulating level of S1P in GBM patients [251] might foster monocyte migration from the peripheral blood into the tumor, and TAM formation.

Different studies support an important regulatory role of SphK1 in the activation of the microglia/macrophages by pro-inflammatory factors [252]. Pro-inflammatory factors, such as LPS and transient cerebral ischemia, result in an increase in the expression and activity of SphK1, and consequently in an increase in the production of S1P [148,149,152,252]. S1P signaling in TAMs leads to increased cellular proliferation, and enhanced production of different pro-inflammatory cytokines (such as TNF-α and IL-1β and IL-17) and nitric oxide, and this signaling is inhibited by SphK1 inhibition or gene knockout of this enzyme [149,253]. S1P1–3 receptor subtypes were identified as relevant in S1P signaling in transient cerebral ischemia [253,254,255,256]. In addition, S1P regulates M1/M2 polarization toward brain damage as a pathogenesis of cerebral ischemia. In particular, in activated microglia of post-ischemic brain, M1 polarization occurs mainly through S1P1, and suppressing S1P1 activity increased mRNA levels of M2 polarization markers [257].

Conversely, Hughes et al. [258] found that during acute inflammation S1P is able to suppress the M1 macrophage polarization through S1P1-G_i_-coupled signaling and inhibition of NF-κB. In addition, it has been shown that S1P signaling for M2 polarization occurs mainly via S1P1 and S1P2, which results in the activation of ERK and consequent IL-4 secretion [259]. Once secreted, IL-4 binds to IL-4Rα and IL-2Rγ receptors, induces SOCS1 and suppresses SOCS3 via stat-6 phosphorylation, and leads to M2 macrophage polarization [259]. It emerged that, besides promoting chemotaxis and pro-inflammatory actions, S1P can silence immune response in macrophages, and this effect plays an important role in cancer immune evasion [11,252,260]. Indeed, different studies demonstrated that S1P polarizes macrophages to the M2 phenotype, and increased expression of heme oxygenase-1 was found to play a significant role in this process [259,261,262,263,264].

Notably, S1P recently emerged as key effector molecule also within the GBM microenvironments, important to M2-type polarization and function of TAMs. In particular, a very recent report demonstrated that, after human macrophage activation, the rapid decrease of SphK2 expression and activity is required for inflammatory cytokine production [265]. However, this downregulation of SphK2 is transient, and this enzyme is upregulated later on, to act as an anti-inflammatory mediator in human macrophages, with an inhibitory effect on inflammatory cytokine production [265]. Of interest, SphK2 acts as anti-inflammatory protein in human macrophages by suppressing LPS-mediated NF-κB activation and mitochondrial ROS formation, independently of its enzymatic activity [265]. Notably, in GBM patients the expression of SphK2 correlates with TAM infiltration as well as with the Ki-67 proliferative index of the tumors [266]. From these pieces of evidence, it is tempting to speculate that S1P in the GBM microenvironment initially recruits microglia/macrophages and then upregulates SphK2 in these cells. This results in the changing of these monocytes from the pro-inflammatory state to the anti-inflammatory one, favoring GBM proliferation and protecting the tumor from an immune intervention.

An interesting study by Luo et al. [267] recently reported that dying cells effectively secrete S1P, which acts as a ‘find-me signal’ for macrophage recruitment. The action of S1P finally resulted in enhanced dead cell phagocytosis and immune tolerance, and occurred through activating erythropoietin signaling in macrophages [267]. Whether this mechanism operates in cancer and in perinecrotic regions of GBM, and contributes to its immune evasion, remains unknown.

The GBM microenvironment includes a further type of monocytic immune cell that participates to immunosuppression, represented by myeloid-derived suppressor cells (MDSCs). These cells account for a major subpopulation of monocytes in the blood of GBM patients, and are abundant in the GBM microenvironments [268,269]. It was demonstrated that S1P can induce an increase in MDSC activity in different solid cancers [270], and thus reduce the immune response. Indeed, the S1P activation of the ERK1/2 MAPK pathway coupled to S1P3 induced an increased expression of granulocyte-macrophage colony-stimulating factor (GM-CSF), resulting in an enrichment of MDSCs in the tumor niche, and autocrine stimulation of immunosuppressive functions of these cells [270]. These results show that S1P3 activation by S1P can stimulate MDSC immunosuppressive activity in the tumor niche. However, currently, there are no studies describing the role of S1P in MDSCs of GBM.

Among immune dysfunctions of cancer patients involved in tumor immune evasion, there is also that of T cells, T-cell lymphopenia being particularly severe in patients with GBM [271]. S1P is increasingly recognized for its role in mediating naïve T-cell egress from lymphoid organs, a process that is dependent on both the high S1P chemotactic gradient between lymphoid organs and blood, and on S1P binding to its S1P1 receptor on the T-cell surface [272,273]. Both processes are altered in GBM patients. Indeed, it was shown that the circulating level of S1P in platelet-rich plasma is significantly reduced in GBM patients [251], reducing the chemotactic gradient that directs T-cell egress into the circulation [32]. Interestingly, a recent study uncovered that T-cell lymphopenia in GBM patients is associated with T-cell trapping into the bone marrow, and to the GBM-induced internalization of the S1P1 receptor from the T-cell surface, which impairs T-cell egress from the bone marrow [274].

### 7.5. S1P in the Cancer Microenvironment Promotes Intense Angiogenesis

GBM exhibits extremely high vascularity, and it is among the most angiogenic tumor in humans, with sustained endothelial activation significantly contributing to its microenvironment and progression [77]. Notably, angiogenesis represents a key process in GBM, associated with uncontrolled cell growth and spread as well as therapy resistance, and, not least, high death rate of patients [39,275]. In addition, the perivascular niche represents a reservoir for GSCs, which promotes tumor progression and drives aggressive behavior [276,277,278]. The major trigger of angiogenesis in GBM is hypoxia, which promotes adaptation through HIFs, a transcription factor involved in the regulation of multiple genes [279]. Among HIF-induced angiogenic factors, VEGF plays a major role in GBM, as it boosts proliferation, permeability, migration, and survival in ECs, and formation of immature, highly permeable blood vessels [280,281]. Indeed, VEGF is thought to be the major angiogenic mediator in GBM [282], and its overexpression in GBM is significantly associated with poor patient survival [283]. In hypoxic GBM cells, SphK1 was shown to upregulate HIF-1α by stabilizing it through the Akt pathway, and by reducing its proteasomal degradation [284]. Additionally, HIF-2α upregulates SphK1 in GBM cells to promote neovascularization [112], and GSCs play a pivotal role in inducing angiogenesis via HIF-1/VEGF [285].

Different studies underscore the role of S1P in ECs. It was reported that S1P is capable of acting as a potent angiogenic factor with potency similar to VEGF, regulating both early and later stages of angiogenesis [286], and to promote neovascularization of tumors [98]. S1P promotes endothelial migration and proliferation, stimulates EC entubulation, and stabilizes newly formed vessels [287,288]. In ECs, S1P upregulates both mRNA and protein levels of the C2H2-zinc finger gene ZNF580, via p38 MAPK pathway [289], and S1P1 is involved in the S1P actions for migration, vascular maturation, and effective induction of angiogenesis [290,291]. In the context of GBM, S1P was shown to initiate microvascular EC sprouting, and SphK1 plays an essential role in regulating paracrine angiogenesis by inducing an increase in both number and length of sprouts [50,96]. This effect was not affected by VEGF, suggesting that S1P signaling is required even when potent angiogenic factors such as VEGF are present. As different pro-angiogenic growth factors and cytokines, S1P and VEGF collaborate in GBM angiogenesis, often mutually reinforcing their action [9,112,289,292]. We recently reported that S1P acts as a potent stimulator of both migration and angiogenesis in ECs derived from different GBM patients too, and S1P1/S1P3 receptors are involved in these effects [96].

### 7.6. S1P in the Cancer Microenvironment Promotes Deregulated Energy Metabolism

Among cancer hallmarks, reprogramming energy metabolism has emerged as critical, in order to adapt to the increased nutritional requirements during the growth, division, and survival of different cancer cells, including GBM ones [166]. Multiple studies strongly support that cancers undergo metabolic adaptation and reprogramming, with factors intrinsic to cancer cells such as oncogenic mutations, and cell-extrinsic microenvironmental factors substantially contributing to the metabolic phenotype of cancer cells [293]. GBM cells demonstrate a striking ability to rewire their metabolism, the TME contributing immensely to their metabolic reprogramming [294,295]. A key event is the increased glycolysis under aerobic conditions (the Warburg effect), which promotes cancer cell proliferation and survival. As in other malignant tumors, GBM cells exhibit high levels of non-oxidative metabolism of glucose even in the presence of oxygen [296], and the upregulation of glycolysis appears to be a universal feature in both GBM and GSCs [297]. The hypoxic microenvironment within the tumor is a major driver of the metabolic shift toward glycolysis [298], which is crucial in providing not only ATP but also different intermediates for anabolic metabolism, as well as adaptation to hypoxic conditions [299,300,301,302].

It is only in the last years that S1P has emerged as a mediator of the deregulated metabolism in cancer cells. In different solid cancers, downregulation/inhibition of SphK1 inhibits the Warburg effect [303,304], suggesting that SphK1 overexpression in GBM contributes to efficient glycolysis. Interestingly, a S1P3-dependent regulatory network was recently identified as a modulator of the Warburg effect during osteosarcoma growth [305]. Moreover, S1P3-signaling activation by S1P inhibits the phosphorylation of Yes-associated protein (YAP), promoting YAP nuclear translocation, formation of the YAP-c-MYC complex, and finally enhancing the transcription of glycolytic enzymes [305].

The metabolic reprogramming of GBM cells is the result of a complex network of mechanisms that, through the activation of oncogenes, induces an increased expression of cell transporters and glycolytic enzymes. Among these oncogenes, HIFα activation by hypoxia is crucial to promote anaerobic glycolysis, to reduce mitochondrial respiration, and to enhance the expression of VEGF in GBM [306]. Intriguingly, in hypoxic GBM cells, SphK1 upregulates HIF-1α [284] and stimulates the production and secretion of VEGF [181], suggesting that S1P acts as a signal involved in GBM metabolic rewiring.

Among intermediates of glucose metabolism in GBM cells, cytosolic glucose-6-phosphate (G6P) functions as negative regulator of the intracellular signaling and invasive phenotype, and its transfer into the ER through the microsomal G6P translocase is highly efficient in GBM cells [210]. Interestingly, S1P-induced ERK phosphorylation and migration of GBM cells was inhibited after treatment with chlorogenic acid, a G6P translocase inhibitor [210], suggesting that cytosolic G6P acts as an inhibitor of S1P signaling. On this basis, it can be speculated that the microsomal segregation of G6P, as well as the low G6P levels in GBM [295], prompts S1P signaling.

Despite these novel pieces of evidence suggesting a role of S1P in tumor metabolic aberrations, further studies are needed to explore this less-studied aspect of S1P signaling. In this context, it is worth mentioning that, among different metabolic deregulations, aberrant lipid metabolism has emerged as crucial, not only as an energy source and in providing substrates for membrane synthesis, but also for its role in cellular signaling in both GBM cells and GSCs [307,308]. Concerning lipid signaling, different studies demonstrated that lysophosphatidic acid (LPA), a major membrane-derived lipid-signaling molecule beside S1P, acts as a key player in GBM [309,310]. LPA is generated extracellularly from lysophosphatidylcholine through the action of autotaxin (ATX), and, similar to S1P, exerts potent cancer-promoting effects through G protein-coupled receptors. Interestingly, ATX can also hydrolyze sphingophosphorylcholine to produce S1P [311], and S1P has been shown to inhibit ATX [312] through regulation of transcription and secretion [313]. Despite these intriguing interconnections between LPA and S1P metabolism, and a cross-talk between these two lysophospholipids in gastric cancer [314], up until now the interaction between S1P and LPA metabolism/signaling in GBM remains unexplored.

## 8. Conclusions

S1P is emerging as a key signal in the reciprocal cross-talk between cancer cells and recruited normal cells in the TME, as well as in their interplay with the TME, complex processes crucial for tumor growth and progression. GBM, one of the most lethal of all human malignancies, is characterized by a unique TME, enriched in S1P, which enables intensive growth and efficient invasion mechanisms, fosters stemness, impairs immune surveillance, promotes angiogenesis, and mediates death resistance. More specifically, cancer cells and GSCs cooperate with resident host cells, including TAMs and ECs, as well as other brain cells, to enrich the TME with S1P. On its turn, S1P acts as a potent tumor-promoter, and supports tumor progression by contributing to all its malignant hallmarks.

The findings reported above, and particularly the impact of S1P in promoting cancer hallmarks (Figure 3), have clear implications for cancer treatment. Indeed, S1P targeting has emerged as a promising therapy for cancer/GBM patients [315]. Significant efforts have been made (and are on track) for the synthesis, identification, and therapeutic evaluation of S1P-targeting drugs in different cancers, including GBM. Many recent investigations, both in cultured cells and in experimental animals, are promising. Among them, promising studies have been carried out using drugs targeting enzymes of S1P metabolism, such as SphKs [65,316,317] and acid ceramidase [318], S1P transporters [319], and S1P receptors and signaling [232,320,321]. Of particular interest is the S1P receptor antagonist FTY720 (Gilenya/Fingolimod), which demonstrated efficacy in multiple sclerosis and has been used in numerous animal experiments and clinical trials for different diseases and cancers, including GBM (Clinicaltrials.gov, NCT02490930, sponsored by Sidney Kimmel Comprehensive Cancer).

Although the targeting of S1P appears an extremely attractive strategy in permitting the inhibition of cancer hallmarks, key issues deserve attention. First, a detailed, comprehensive understanding of S1P regulation (production, extracellular export, and degradation) and of S1P receptor expression/interaction is required for a full understanding of the role of S1P in the TME. Second, crucial to the use of S1P signaling as a target in cancer is elucidation of how the diversity of TMEs contributes to their S1P enrichment and understanding of the S1P-signaling circuits that operate across different cells in the TME. The key to resolving this is an expansion of our mechanistic understanding about the actual role of S1P on various cells in the TME, and which of the diverse effects of S1P are relevant to the maintenance/progression of cancer. Third, despite the requirement for S1P signaling emerging as a crucial mechanism that conspires in tumorigenesis, this signaling appears a basic need for most types of cells to live. Thus, targeting S1P should be well balanced between the agreeable outcome of eradicating malignant cells and the menace afforded to healthy normal cells/tissues. Fourth and last, but not least, the heterogeneity of individual cancers, particularly of GBM, might require patient-specific personalized therapies based on responsiveness to treatments. In this regard, the possibility to develop a patient-specific S1P panel might be of help in selecting S1P-based target therapies. Going further with this knowledge should help close the gaps in our understanding of the role of S1P in GBM/cancer progression, and pave the way for the identification of target key modules in S1P signaling with maximum potency and selectivity for cancer. Although the challenges are still substantial, past and recent advancements support the idea that further breakthroughs will occur in the near future, with impact.

## Figures and Tables

**Figure 1 cells-09-00337-f001:**
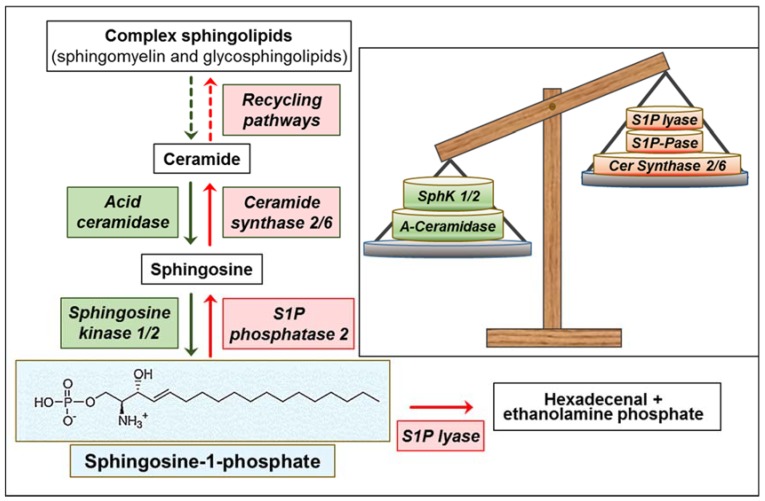
Overview of sphingosine-1-phosphate (S1P) metabolism and its alterations in glioblastoma (GBM). Green: overexpressed/upregulated enzymes; red: downregulated enzymes. Green and red arrows, increased and decreased enzyme activity, respectively. The insert shows the imbalance between enzymes involved in S1P formation (green) and degradation (red).

**Figure 2 cells-09-00337-f002:**
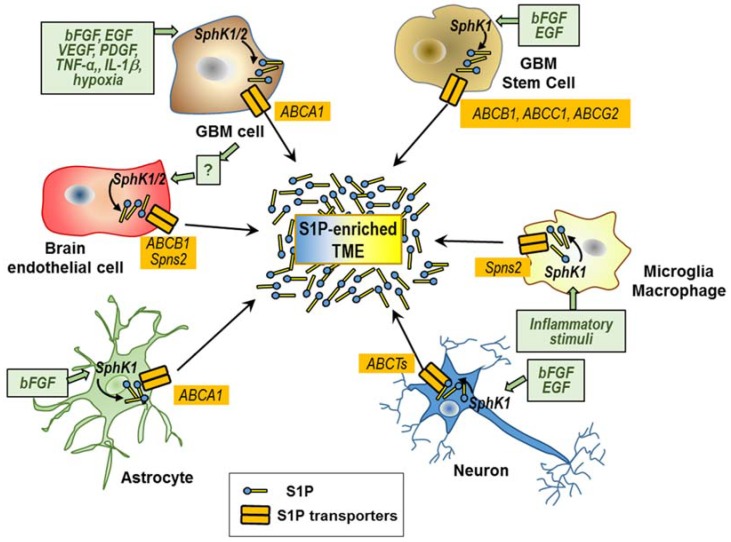
Cellular types involved in the S1P enrichment of the tumor microenvironment. Stimuli (green boxes) of S1P synthesis and export in different cells of the GBM microenvironment are shown. Putative S1P transporters are shown in yellow boxes.

**Figure 3 cells-09-00337-f003:**
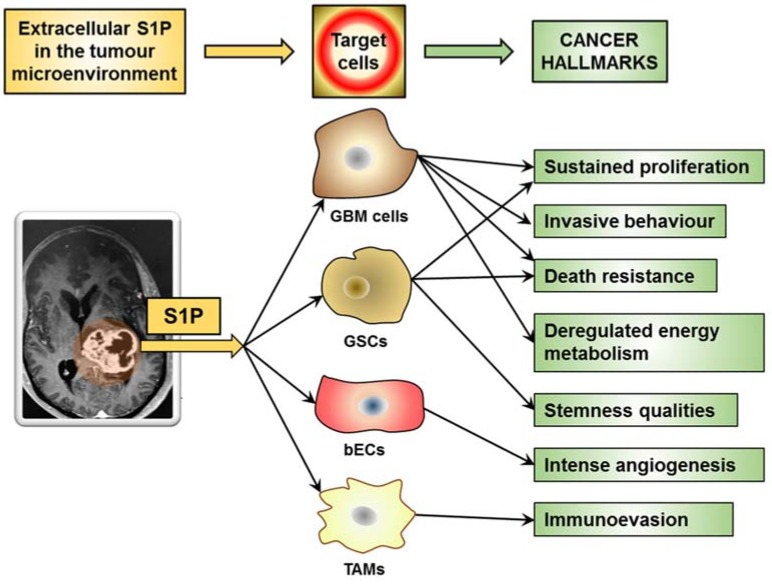
S1P in the GBM microenvironment promotes aggressive cancer hallmarks.

**Figure 4 cells-09-00337-f004:**
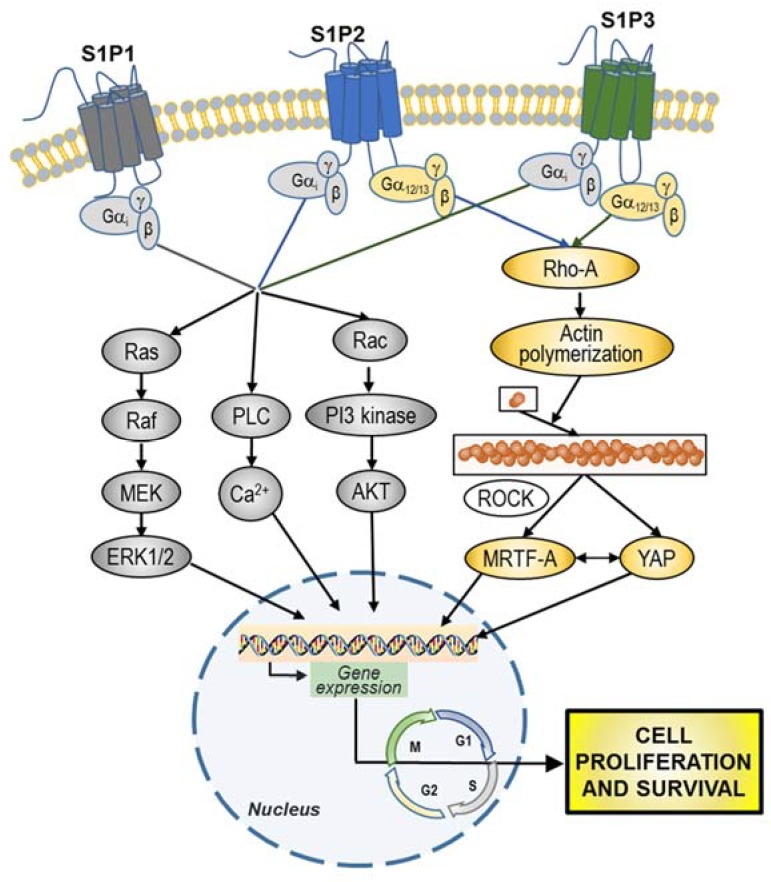
Overview of S1P-signaling pathways involved in GBM cell and GBM stem cells (GSC) proliferation and survival. S1P, through stimulation of i) G_i_ may simultaneously activate MAPK-ERK1/2, phospholipase C (PLC), and phosphoinositide 3-kinase (PI3K) pathways; and ii) G_12/13_ may promote actin polymerization through RhoA. Following the activation of downstream signaling pathways, S1P prompts the activation of different transcription factors, leading to the regulation of different genes involved in the promotion of cell proliferation and survival.

**Figure 5 cells-09-00337-f005:**
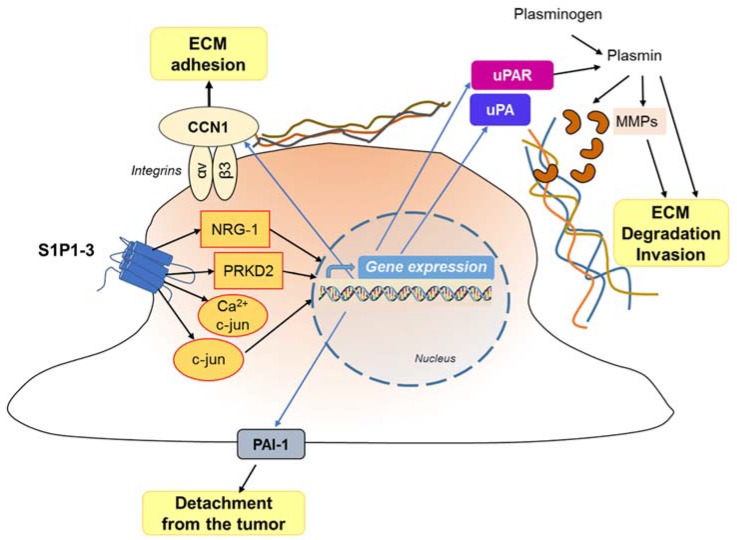
Effect of S1P on GBM invasivity. The interaction of S1P with its receptors results in the activation of different signaling pathways, and enhanced gene and protein expression, leading to detachment (at the trailing edge) and attachment (at the leading edge) of the migrating cell, with stimulation of the extracellular matrix degradation favoring and cell invasion.

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
