# Peer review of "Sphingosine-1-Phosphate in the Tumor Microenvironment: A Signaling Hub Regulating Cancer Hallmarks"

_cells, 2020, doi:10.3390/cells9020337_

Round 1

Reviewer 1 Report

Here the authors reviewed and discussed all the evidence existing in terms of the role of S1P signaling in glioblastoma. The review is very complete and detailed, in agreement with the experience of the authors in the topic.

The review focuses mostly on the role of S1P in glioblastoma as is the most studied scenario. However, it would be very interesting to mention briefly if there is similar evidence in other brain cancers. What would make so specific the S1P-signaling for glioblastoma? Is there any similar evidence for another type of tumors?

The authors have gathered the current information existing on S1P metabolism. However, lipid metabolism is highly interconnected. Hence, is there any other alteration on lipid metabolism worth mentioning in glioblastoma?

Minor comment: It seems that the definition of "Khps1" (line 153) is missing.

Author Response

"The review focuses mostly on the role of S1P in glioblastoma as is the most studied scenario. However, it would be very interesting to mention briefly if there is similar evidence in other brain cancers..."

We thank the Reviewer for his/her constructive comments. We enriched the review by adding the actual knowledge on these topics, and discussed it (page 7, first paragraph).

"The authors have gathered the current information existing on S1P metabolism. However lipid metabolism is highly interconnected. Hence, is there any other alteration on lipid metabolism..."

This is an interesting issue, which would require a dedicated review. Taking into consideration the Reviewer suggestion, and the importance of the topic, we added to the text a short description of the current knowledge on lipid metabolism in GBM, and we detailed the connections/cross-talks between S1P  and lysophosphatidic acid in this cancer, including the related references and a comment (page 27, 2nd paragraph).

"Minor comment; It seems that the definition of "Khps1" is missing."

This name was given to the long non coding RNA that is transcribed in antisense orientation to the SphK1. This clarification has been added to the text.

Reviewer 2 Report

This work is well organized and comprehensively described the role of sphingosine-1-phosphate (S1P) in glioblastoma. In consequence, this manuscript will contribute to the understanding the participation of S1P in the tumor microenvironment.

Minor opinion:

1. The mechanisms underlying the pro-survival properties of S1P mainly include signalling

pathways that result in the inhibition of apoptosis and/or the induction of protective autophagy

(Figure 4). (Page 15, lines 620-622)

Neither apoptosis nor autophagy is described in Figure 4.

2. Please delete space between ‘not only’ and ‘a greater’. (Page 17, line 686)

Author Response

"This work is well organized and comprehensively described the role pf sphingosine-1-phosphate..."

We wish to thank the Reviewer for appreciating our efforts in writing the review in a well organized and comprehensive fashion.

Minor opinion:

1. "The mechanisms underlying..."

Thank you for your attention. The insertion of Figure 4 here was a mistake. This was eliminated and substituted with proper references. 

2. "Please delete space..."

Done